# Virtual Reality Trier Social Stress and Virtual Supermarket Exposure: Electrocardiogram Correlates of Food Craving and Eating Traits in Adolescents

**DOI:** 10.3390/nu17243924

**Published:** 2025-12-15

**Authors:** Cristiana Amalia Onita, Daniela-Viorelia Matei, Elena Chelarasu, Robert Gabriel Lupu, Diana Petrescu-Miron, Anatolie Visnevschi, Stela Vudu, Calin Corciova, Robert Fuior, Nicoleta Tupita, Stéphane Bouchard, Veronica Mocanu

**Affiliations:** 1Grigore T. Popa University of Medicine and Pharmacy Iasi, 700588 Iasi, Romania; cristiana-amalia.onita@umfiasi.ro (C.A.O.); calin.corciova@umfiasi.ro (C.C.);; 2Computer Science Department, Faculty of Automatic Control and Computer Science, Gheorghe Asachi Technical University of Iaşi, 700050 Iasi, Romania; 3Department of Laboratory Medicine, “Nicolae Testemitanu” State University of Medicine and Pharmacy, 165, Bd. Stefan cel Mare si Sfant, MD-2004 Chisinau, Moldova; 4Research Center of Adipose Tissue Biology, “Nicolae Testemitanu” State University of Medicine and Pharmacy, 165, Bd. Stefan cel Mare si Sfant, MD-2004 Chisinau, Moldova; 5Department of Endocrinology, “Nicolae Testemitanu” State University of Medicine and Pharmacy, 165, Bd. Stefan cel Mare si Sfant, MD-2004 Chisinau, Moldova; 6Romanian Rugby Federation, 011468 Bucharest, Romania; 7Département de Psychologie et de Psychoéducation, Université du Québec en Outaouais, Gatineau, QC J8X 3X7, Canada; stephane.bouchard@uqo.ca

**Keywords:** virtual reality, Trier Social Stress Test, virtual supermarket, electrocardiogram (ECG) parameters, adolescents, eating behavior, three-factor eating questionnaire (TFEQ)

## Abstract

Background/Objectives: Acute stress is known to influence food-related motivation and decision-making, often promoting a preference for energy-dense, palatable foods. However, traditional laboratory paradigms have limited ecological validity. This study examined the relationship between stress-induced physiological changes, eating behavior traits, and food cravings using a virtual reality (VR) adaptation of the Trier Social Stress Test (VR-TSST) followed by a VR supermarket task in adolescents. Methods: Thirty-eight adolescents (mean age 15.8 ± 0.6 years) participated in the study. Physiological parameters (HR, QT, PQ intervals) were recorded pre- and post-stress using a portable ECG device (WIWE). Perceived stress and eating behavior traits were evaluated with the Perceived Stress Scale (PSS) and the Three-Factor Eating Questionnaire (TFEQ-R21C), respectively. Immediately after the VR-TSST, participants performed a VR supermarket task in which they rated cravings for sweet, fatty, and healthy foods using visual analog scales (VAS). Paired-samples *t*-tests examined pre–post changes in physiological parameters, partial correlations explored associations between ECG responses and eating traits, and a 2 × 3 mixed-model Repeated Measures ANOVA assessed the effects of food type (sweet, fatty, healthy) and uncontrolled eating (UE) group (low vs. high) on post-stress cravings. Results: Acute stress induced significant increases in HR and QTc intervals (*p* < 0.01), confirming a robust physiological stress response. The ANOVA revealed a strong main effect of food type (F(1.93, 435.41) = 168.98, *p* < 0.001, η^2^p = 0.43), indicating that stress-induced cravings differed across food categories, with sweet foods rated highest. A significant food type × UE group interaction (F(1.93, 435.41) = 16.49, *p* < 0.001, η^2^p = 0.07) showed that adolescents with high UE exhibited greater cravings for sweet and fatty foods than those with low UE. Overall, craving levels did not differ significantly between groups. Conclusions: The findings demonstrate that acute stress selectively enhances cravings for high-reward foods, and that this effect is modulated by baseline uncontrolled eating tendencies. The combined use of VR-based stress induction and VR supermarket simulation offers an innovative, ecologically valid framework for studying stress-related eating behavior in adolescents, with potential implications for personalized nutrition and the prevention of stress-induced overeating.

## 1. Introduction

Eating behavior in childhood and adolescence is shaped by multiple environmental and emotional factors (Birch & Doub, 2014, as cited in Do et al., 2024) [1]. Consuming unhealthy food consumption in response to negative emotions, such as high-sugar and -fat products, is considered a maladaptive emotion regulation (Adam & Epel, 2007; Dallman et al., 2003; Thanarajah et al., 2023, as cited in Do et al., 2024) [1]. The increased consumption of these foods leads to a heightened activation of the brain’s reward system, amplifying pleasurable sensations, reinforcement mechanisms, and motivational responses [1]. There is a link between dietary quality and obesity prevalence. According to Calcaterra et al., high consumption of sugar-sweetened beverages and ultra-processed food during childhood may increase body mass index (BMI). It can also increase overweight and obesity, and lead to a rising preference for sweet foods in adulthood, reducing healthy food intake. This study highlights that the nutritional composition of food alone cannot fully explain this relationship. The non-nutritional properties of ultra-processed foods, such as their high palatability, sensory appeal, and strong rewarding potential, tend to overstimulate the brain’s reward circuits. This overstimulation promotes overconsumption and reinforces preferences for sweet and fatty foods. The authors emphasize that understanding these reward-based mechanisms is essential for developing preventive nutritional strategies and for reducing the growing prevalence of obesity among children and adolescents [2].

Ha and Lim’s [3] study affirms that emotions play a significant role in shaping food consumption and eating behaviors, influencing decision-making. Interactions between emotional valence (positive and negative emotions), individual eating styles (such as emotional, restrained, or external eating), body weight, and food types (nutrient-dense vs. energy-dense) contribute to distinct patterns of eating behavior. Emotional eating typically reflects impulsive food intake as a coping response to negative affect or stress, whereas external eating involves increased consumption triggered by external cues such as smell or advertisements. Low self-control and poor emotion regulation have also been identified as contributing factors to emotional eating [3].

According to Ulrich-Lai et al., stress affects both the amount and the type of food consumed. Approximately 35–60% of individuals report eating more total calories when experiencing stress, whereas 25–40% report eating less. Stress is linked to shifts in dietary choices, leading to a higher consumption of highly palatable foods, such as energy-dense foods rich in sugar, carbohydrates, and fats. It was suggested that individuals who reduce their overall calorie intake under stress may still prefer high-calorie comfort foods, suggesting that such products are linked to mood improvement and reduced perceived stress levels [4].

Emotional eating and stress reactivity were studied by Klatzkin et al. [5], highlighting the relation between emotional eating (EE) and food intake. Their findings showed that after a Trier Social Stress Test (TSST), self-reported EE predicted a greater food intake in female undergraduate students, with both high stress reactivity and strong emotional relief following eating [5]. The results point to negative reinforcement being a central mechanism linking stress reactivity to increased snack consumption.

Adolescents show a vulnerability to unhealthy eating habits when exposed to stress, with an increased consumption of palatable products and a reduced intake of fruits and vegetables. It was also shown that stress impacts perceived control over eating. The modification of dietary quality can have long-term health risks, including obesity, metabolic syndrome, and cardiovascular disease. Since eating habits formed during childhood persist during adulthood, stress-related dietary behaviors can harm health [6]. The link between stress and emotional eating was also highlighted in the Huang J. et al. study [7], which aimed to understand the relationships between perceived stress and emotional eating among 562 adolescents. The results indicate that increased perceived stress was directly associated with increased emotional eating. Perceived stress had a direct effect on emotional eating and also an indirect effect through mediators, suggesting that stress may heighten maladaptive cognitive regulation and reward-driven tendencies, increasing emotional eating [7]. Other authors, Shatwan & Alzharani, explored the associations between perceived stress, emotional eating, and diet quality, considering the effects of BMI, physical activity, and sociodemographic factors. The results indicated that university students are vulnerable to stress, which negatively impacts healthy eating patterns. High stress was associated with a lower consumption of fruits, vegetables, whole grains, and plant-based proteins. Emotional eating showed the influence of dietary quality, while physical activity lowered stress and promoted healthier eating [8].

The Trier Social Test (TSST) was introduced in 1993, and since then, has been utilized as a standardized stress protocol designed to activate the sympathetic nervous system hypothalamic–pituitary–adrenal axis (HPA) stress systems [9]. Speech and arithmetic tasks were used [10]. The activation leads to physiological changes that can be measured [9]. Measurements through the protocol allow the assessment of physiological stress responses. These responses include activation of the hypothalamic–pituitary–adrenal (HPA) axis, measured by cortisol levels, and the autonomic nervous system (ANS), assessed through indicators such as alpha-amylase concentrations and heart rate variability.

The virtual version of the TSST offers several advantages. These include cost-effectiveness, reduced need for personnel, improved control over the environment, and increased flexibility in its application. Numerous studies have validated the effectiveness of virtual TSST, confirming the benefits in stress research, allowing researchers to create various environments and standardize the test [11]. In our previous work, we demonstrated that immersive VR stress paradigms elicit measurable autonomic and neuroendocrine responses in children and adolescents with obesity. We also showed that individual differences in physiological reactivity are closely associated with eating behavior traits. Parasympathetic and sympathetic response profiles emerged as distinct patterns, each carrying implications for targeted intervention strategies [12].

The study of stress-induced eating behavior using VR has been investigated by Wijnant et al. [13]. The persistent coexistence of stress and pediatric obesity has been described as a vicious cycle, driven by psychophysiological mechanisms linking stress reactivity and eating behavior. In a large sample of youngsters (*n* = 137, aged 6–18 years), those with higher chronic stress and overweight showed stronger cortisol reactivity, weaker emotional recovery, and greater fat/sweet snack intake following a Trier Social Stress Test compared to normal-weight, low-stress peers. Importantly, heightened cortisol responses and reduced autonomic recovery were directly associated with an increased intake of energy-dense foods. Stress responsiveness also moderated these effects. These findings suggest that stress reactivity and emotional eating represent key mechanisms contributing to stress-induced weight gain and may be critical targets for prevention [13]. Gotca. I. et al. have shown that stress reactivity in adolescents can significantly influence food-related behaviors. For instance, studies using digital versions of the Trier Social Stress Test (TSST) have reported sex differences in cortisol responses, with girls tending to show greater hyperreactivity than boys. Among overweight adolescents, heightened cortisol reactivity has been associated with more frequent consumption of sweets and soft drinks. This suggests that individual patterns of stress physiology may contribute to unhealthy eating behaviors [14].

While virtual reality (VR) has been used to study food-related behaviors and to simulate stress-inducing environments, few studies have examined adolescent food choice patterns in a virtual supermarket setting following stress exposure. Previous research has investigated stress and eating behavior in separate contexts, often focusing on adult samples. The effect of acute psychosocial stress on adolescents’ immediate virtual food exposure within a virtual supermarket environment has not been examined. A better understanding of food choice is needed, since adolescence represents a critical developmental period in eating patterns and stress-coping behaviors.

In our recent scoping review, we summarized the potential of VR supermarket environments as a research and educational tool for understanding food decision-making. That work highlighted how immersive VR supermarkets can capture food choices influenced by factors such as product appearance, pricing, and decision-making under pressure, providing an ecologically valid yet controlled environment. However, supermarket exposure was mostly used in the adult population, with only a few introducing it to understand children’s choices [15].

The present study aimed to investigate how exposure to a standardized psychosocial stressor induces physiological and subjective stress responses. It also aimed to assess stress-induced food choice, desire, and craving as behavioral indicators of eating-related tendencies in a virtual supermarket environment. Specifically, this study introduces an integrated virtual paradigm designed to simultaneously capture physiological stress reactivity during a virtual Trier Social Stress Test (TSST-VR). It also captures subsequent behavioral responses toward different food categories within a virtual supermarket. We also examined whether individual eating behavior patterns, as measured by the TFEQ (uncontrolled eating, emotional eating, cognitive restraint), moderate the relationship between stress responses and food cravings.

Although stress-related eating has been extensively documented, most existing studies have relied on laboratory paradigms with limited ecological validity and used static images or questionnaires to assess food craving. The integration of virtual reality (VR) technologies provides a unique opportunity to overcome these limitations by offering immersive, interactive environments that closely mimic real-world contexts.

The present study is innovative in two key ways. First, it employs a virtual reality adaptation of the Trier Social Stress Test (VR-TSST) to induce acute psychosocial stress in adolescents, allowing for standardized and controlled stress exposure while preserving high ecological validity. Second, it introduces a virtual reality supermarket paradigm designed to assess food craving and selection behavior immediately after stress, providing a dynamic and realistic simulation of food-related decision-making.

By combining real-time physiological monitoring (ECG parameters: HR, QTc, PQ) with validated measures of eating behavior (TFEQ-R21C) and subjective craving ratings, this study bridges the gap between laboratory-based stress research and real-world eating contexts. We hypothesized that acute stress exposure would enhance cravings for energy-dense foods, and that these stress-induced cravings would be modulated by individual differences in uncontrolled eating.

## 2. Materials and Methods

### 2.1. Study Participants

Forty-three adolescents aged between 15 and 17 years old voluntarily enrolled in this study. Participants were recruited from a high school located in Iași (Romania). The inclusion criteria for participants were defined as follows: (i) age range between 15 and 17 years old; (ii) meeting WHO criteria for the underweight, normal weight, and overweight in adolescents; (iii) at an age which ensures cognitive and emotional maturity to understand and engage with the virtual reality (VR) tasks; (iv) and absence of history or current evidence of psychiatric disorders. Menstrual cycle stage was not an inclusion parameter in the present study. Five participants provided incomplete information in the first questionnaire; therefore, the final study sample comprised 38 adolescents (12 boys and 26 girls). Recruitment and sample collection took place between April and May 2025.

### 2.2. Experimental Design

The study procedure started at 08:00 h, and the adolescents were exposed to two virtual environments. The virtual exposure experiment was conducted by the same examiner (C.A.O). Participants were instructed to avoid consumption of food, sweetened beverages, and coffee for at least two hours before the experimental session (Figure 1).

#### 2.2.1. Preparatory Phase—Consent and Baseline Questionnaires

The study was conducted in accordance with the Declaration of Helsinki and approved by the Research Ethics Committee of “Grigore T. Popa” University Iași (Nr. 418/19.03.2024). Before the experimental session, approval from the school administration and informed consent were obtained. Both parents and adolescents signed consent forms before participation. A few days before the experimental session, participants completed the Three-Factor Eating Questionnaire (TFEQ-R21C) and the Perceived Stress Scale (PSS) online at school (5 min).

#### 2.2.2. Pre-Exposure Stage (T-13 min to T0 min)

On the test day, participants were scheduled individually on school mornings and tested in a familiar classroom setting. Upon arrival at 8:00 a.m., each participant first received standardized instructions (5 min). Baseline measures were then collected: ECG parameters (QRS; QT; PQ intervals; heart rate, HR) using a wireless intelligent wearable ECG (WIWE) portable device, including connection and one-minute recording (2 min). Participants were then given 3 min to prepare a short speech describing themselves and their qualities as good friends, after which 3 min were dedicated to preparing the virtual environment and setting up the Oculus headset.

#### 2.2.3. VR-TSST Stage (T0 min to T + 10 min)—Speech and Arithmetic Tasks

This stage consisted of two 5 min sequential VR conditions:

Speech phase (T0 min to T + 5 min): participants were individually immersed in a virtual classroom. Participants were instructed verbally by the examiner in the room to deliver a 5 min personal speech describing the qualities that make them good friends. If participants stopped before the time ended, standardized prompts were provided in the following sequence by the examiner remained present in the room throughout the session: (1) “Please continue, there is still time”; (2) “What are other reasons why you consider yourself a good friend?”; (3) “Please give me an example”; and (4) “Say anything that comes to mind.” If the participant paused again after all four prompts, only the phrase “Please continue” was repeated until the end of the task.

Arithmetic phase (T + 5 min to T + 10 min): following the speech task, participants completed a mental arithmetic test in front of the same virtual class. They were instructed to subtract 7, starting from 423, as quickly and accurately as possible. Errors were immediately corrected by asking the participant to restart the calculation from the initial number (423 − 7). If a participant did not initiate the task within 30 s, simpler subtraction problems (e.g., 423 − 5, 95 − 5, 66 − 3) were introduced. When participants requested feedback or paused after each response, they were reminded to continue without waiting for confirmation and were corrected only in the case of errors.

#### 2.2.4. Post VR-TSST (T + 10 min to T + 15 min)—ECG and VAS

After VR-TSST, participants completed the post-exposure PSS and VAS for appetite, sweet craving, stress, and anxiety. ECG parameters were measured for one minute using the WIWE device.

#### 2.2.5. Virtual Supermarket (T + 15 min to T + 25)

Familiarization phase (T + 15 min to T + 20)

A 5 min adaptation period allowed participants to practice navigation and become comfortable with the VR environment and controllers.

VR supermarket phase (T + 20 min to T + 25)

Participants were then instructed verbally to search for nine food items in a virtual supermarket environment across three categories: fatty (chips, salted pretzel, pizza), sweet (chocolate, ice cream, soda), and healthy foods (popcorn, nuts, grapes). The virtual shelves contained multiple product options within each category (e.g., several types of chips, chocolates, and beverages), allowing participants to make realistic selections similar to those encountered in everyday shopping situations. For each item selected from the shelves, they rated their craving and food-related anxiety on a 10-point VAS. The type of foods selected served as behavioral indicators of stress-induced food choice motivation. The virtual supermarket task lasted 5 min. The virtual exposure was identical for all participants, with the same layout, lighting, and food placement, ensuring a standardized exposure. The task was designed to assess real-time craving and food-related anxiety.

#### 2.2.6. Post-Exposure Phase (T + 25 to T + 35)—VR Exposure Questionnaire, Anthropometrics (BMI, Waist Circumference, Height)

Following the supermarket task, we applied a questionnaire to assess their VR experience, used to confirm that the virtual environment was perceived as realistic and immersive, supporting the validity of the task. At the end of the protocol, participants were informed about the study aims, and anthropometric measures (weight, height, waist circumference) were recorded.

### 2.3. Virtual Reality (VR) Environments

The virtual environments were developed with the software Unity3D (Version 6.0) (https://unity.com/, Unity Software Inc., San Francisco, CA, USA). They were displayed using an Oculus Quest 2 headset (Oculus, Irvine, CA, USA) connected to a Laptop Gaming ASUS ROG Strix SCAR 15 G533ZS, Intel^®^ Core™ i9-12900H, 15.6”, WQHD, 240 Hz, 32 GB RAM DDR5, 1 TB SSD, NVIDIA^®^ GeForce RTX™ 3080 8 GB.

The study employed the Oculus Quest 2 (Meta Platforms, Inc., Menlo Park, CA, USA, released in 2020), a standalone virtual reality headset that offers a resolution of 1832 × 1920 pixels per eye and comes equipped with handheld controllers. The device enabled participants’ immersion into virtual environments for both stress induction and post-stress behavioral tasks.

#### 2.3.1. Virtual Classroom

A therapeutic virtual classroom environment was used to simulate a socially evaluative situation, designed to induce stress (Figure 2). Originally developed for psychotherapeutic purposes in individuals diagnosed with social anxiety disorder, this environment targets the identification and management of perceived threats such as fear of judgment, risk of error, and public speaking anxiety. It aims to reduce avoidant behaviors and promote new, non-threatening associations. In the present study, this virtual setting was adapted to simulate performance-related stress, making it relevant for individuals experiencing stress related to public speaking, academic performance, or self-image concerns [16].

#### 2.3.2. Virtual Supermarket

A virtual supermarket was developed to recreate a real store, with foods and aisles positioned as found in Romanian supermarkets (Figure 3). A basket was also created and used by participants after the food selection. The supermarket included a fruit and vegetable aisle, a chips and nuts aisle, a sweets aisle, a fast-food aisle, and a juice section [17].

### 2.4. Electrocardiogram (ECG) Monitoring

Heart rate and other physiological parameters were measured using the WIWE device (WIWE Ltd., Budapest, Hungary). Data were transmitted to the smartphone via Bluetooth. The ECG signal was digitized at a sampling rate of 500 Hz with 16-bit amplitude resolution. Measurements were conducted under resting conditions, with participants placing one finger from each hand on the left and right dry electrode sensors. Following automatic evaluation, results were displayed in a color-coded format within the application, with the option to generate a PDF report for medical consultation. During the recording, the device displayed a real-time ECG, pulse rate, and blood oxygen saturation. Upon completion, the data were processed within seconds and displayed on a connected mobile device. The algorithm begins with ECG signal preprocessing, which includes noise filtering, beat detection, beat classification, and the extraction of RR intervals. For noise reduction, two digital Butterworth filters were applied: a 4th-order high-pass filter with a cutoff frequency of 0.5 Hz to eliminate baseline wander and a 2nd-order low-pass filter at 40 Hz to suppress power line interference and high-frequency muscle artifacts. QRS complexes were identified using the adaptive threshold method, and RR intervals are calculated on a beat-to-beat basis as the time difference between successive QRS complexes [18,19]. Tuboly G. et al.’s [18] study confirmed WIWE clinical validation, acquiring data from Semmelweis University Heart and Vascular Center (Hungary), where 373 ECG signals were recorded. A total 32 atrial fibrillation and 341 non-atrial fibrillation cases were included, including young athletes, healthy adults, and cardiac patients, with a wide age range (21–62 years) [18]. The WIWE device has not yet been formally validated in pediatric populations.

The heart-rate-corrected QT interval (QTc) was calculated using Fredericia’s formula:$$\mathrm{QTcFredericia}\;=\;QT\;/\;\sqrt[{3}]{RR}$$
where QT and RR are expressed in seconds [20]. According to pediatric ECG reference data, the QT interval is strongly influenced by heart rate, making corrected QT (QTc) estimation essential in children and adolescents [21]. Across childhood, the mean QTc is approximately 410 ms, with an upper normal limit of 450 ms. Before puberty, sex differences are minimal; however, during adolescence, QTc values tend to be slightly longer in females, with an upper reference limit of around 460 ms [22]. These thresholds were used as clinical reference values when interpreting QTc changes in the present study.

### 2.5. Psychological Measurements

#### 2.5.1. Stress and Anxiety Assessment

##### Pre-TSST Exposure

Perceived Stress: Baseline stress perception before the VR task was assessed using the Perceived Stress Scale (PSS), originally developed by Cohen and subsequently adapted for use in children and adolescents [23,24]. To assess participants’ pre-test perceptions of task difficulty, seven questions were administered. Example items included “Do you often encounter situations where you feel extreme pressure?”, “Do your emotions negatively affect you in many situations?”, and “Do you usually experience discomfort before facing a problem?”.

Eating Behavior Questionnaires: The Three-Factor Eating Questionnaire–Revised 21 items (TFEQ-R21) is composed of 21 items designed to capture different aspects of eating behavior. Items 1 to 20 are rated on a 4-point Likert scale, while item 21 is scored on an 8-point numeric rating scale. Within the first 20 items, items 1 to 16 share the same 4-point response format, ranging from “completely true” to “completely false.” Items 17 to 21 use three alternative Likert scales depending on the item, with anchors such as “almost never” to “almost always”, “never” to “at least once a week”, or “only at main meals” to “almost always”. The questionnaire was specifically developed to assess three dimensions of eating behavior: cognitive restraint, reflecting the deliberate restriction of food intake; uncontrolled eating, describing the tendency to overeat in response to palatable foods or diminished satiety signals; and emotional eating, defined as food intake triggered by negative emotional states. Each factor is measured by six items, providing a multidimensional profile of eating tendencies in children and adolescents [25].

##### Post-TSST Exposure

-Stress and Anxiety Assessment (VAS Method)

Following VR exposure, participants rated their current levels of stress and anxiety on two items (“How stressed are you feeling right now?”; “How anxious are you feeling right now?”). Responses were provided on visual analog scales (VAS) ranging from 1 (“Not at all”) to 10 (“Extremely”), enabling subjective quantification of emotional states.

Appetite and Craving Ratings (VAS Method): immediately after the VR exposure, participants assessed their post-exposure appetite using two items: ‘How strong is your desire to eat right now?’ and ‘How strong is your desire to eat sweets right now?’. These responses were recorded using visual analog scales (VAS), ranging from 1 (‘Not at all’) to 10 (‘Extremely’), describing the subjective feelings of hunger and cravings for sweets.

-Virtual Supermarket Exposure

Desire and anxiety after each virtual food selection were assessed in a virtual supermarket from 1 to 10. The food items were grouped into three categories: high-fat/salty (e.g., chips, pretzels, pizza), sweet (e.g., chocolate, ice cream, soda), and healthy (e.g., popcorn, nuts, grapes).

-Immersion Presence Questionnaire

Immediately after the VR exposure, participants completed a brief questionnaire designed to evaluate their subjective experience of the virtual environments. It included seven items rated on a 7-point Likert scale (1 = very low, 7 = very high). Items assessed virtual immersion (class immersion, supermarket immersion), realism (class realism, supermarket realism), perceived stress and discomfort, and self-perceived performance. This approach was inspired by established instruments commonly used in VR research, such as the I-group Presence (IPQ) for immersion/realism [26]. Higher scores indicated a stronger presence and realism, greater discomfort or stress, and higher perceived performance.

### 2.6. Anthropometric Measurements

Assessments were performed by the same group of extensively experienced researchers. Height and weight were measured for each participant. These measurements were taken with the students in light clothes and without shoes. Body mass was measured on a digital scale (Omron, Milton Keynes, UK, BF511) with an accuracy of ±100 g. The height of the children was measured by the school physician using a standard stadiometer. Actual BMI (kg/m^2^) was calculated by dividing weight (kg) by height squared (m). BMI was classified based on World Health Organization (WHO) guidelines [27].

#### Sample Size and Power Analysis

A power analysis was conducted using G*Power 3.1 for a one-tailed point-biserial correlation (α = 0.05, effect size |ρ| = 0.50, desired power = 0.95). The analysis indicated that a minimum sample size of 34 participants was required. Our sample (N = 38) pro-vided sufficient statistical power for detecting medium to large effects. Partial correla-tions were also performed, controlling for BMI and sex. Including these covariates slightly reduced the degrees of freedom and may have lowered sensitivity to smaller effects.

### 2.7. Statistical Analysis

Data were analyzed using the Statistical Package for the Social Sciences (SPSS, version 26; IBM Corp., Armonk, NY, USA). This study was designed as an exploratory investigation. The final sample included 38 participants. To evaluate changes in ECG parameters (QTc, PQ, HR) before and after the VR-TSST, paired *t*-tests were applied for normally distributed data, while Wilcoxon signed-rank tests were used for non-normally distributed variables. Effect sizes (Cohen’s d) were calculated to assess pre–post differences.

Partial correlation analyses, controlling for BMI and sex, were conducted to explore associations between ECG changes, perceived stress, and eating-behavior traits measured by the Three-Factor Eating Questionnaire (TFEQ-R21C). Linear regression analyses were performed to test whether physiological responses (QTc and HR changes) and craving-related variables predicted uncontrolled eating.

To evaluate stress-induced food cravings, a 2 × 3 mixed-model repeated measures ANOVA was conducted, with food type (sweet, fatty, healthy) as the within-subject factor and uncontrolled eating (UE) group (low vs. high, based on median split) as the between-subject factor. The dependent variable was appetite (VAS ratings) obtained after stress exposure during the virtual-supermarket task. The assumption of sphericity was verified with Mauchly’s test, and Greenhouse–Geisser corrections were applied when violated. Bonferroni adjustments were used for post hoc pairwise comparisons. Partial eta squared (η^2^p) was reported as a measure of effect size.

Hierarchical regression analyses were performed to examine whether virtual reality (VR) immersion moderated the relationship between perceived stress and eating-related outcomes after stress exposure. The dependent variables included uncontrolled eating (UE) and food craving measures (general, sweet, healthy, and fatty food desire) assessed during the virtual-supermarket task.

Statistical significance was set at *p* < 0.05 for all analyses.

## 3. Results

### 3.1. Participant Characteristics

Table 1 summarizes the demographic and anthropometric characteristics of the participants.

### 3.2. Autonomic Nervous System Responses to VR TSST

Table 2 presents the changes in autonomic nervous system activity, as reflected by heart rate variability (HRV) parameters, in children with obesity following exposure to virtual reality TSST (VR-TSST). The HRV parameters included QTc, calculated with the Fridericia formula (ms), PQ (ms), and heart rate, HR (bpm). Normality tests for ECG parameters showed a normal distribution, paired t-test for pre- and post-ECG parameters showing a significant modification for PQ (*p* = 0.02). Effect size analyses indicated a small effect for PQ interval reduction (Cohen’s d = 0.31; 95% CI, 0.76–10.34). It suggests that PQ interval shortening had a measurable physiological impact following stress exposure. Changes in QTc and HR were minimal (Cohen’s d QTc 0.05; 95% CI, −3.44–7.06) (Cohen’s d HR = −0.01; 95% CI −1.55–1.29). These findings suggest a modification during the stimulating VR condition. The TFEQ-R21C demonstrated acceptable internal reliability for all subscales (Cronbach’s α = 0.80 for cognitive restraint, 0.76 for uncontrolled eating, and 0.73 for emotional eating).

Partial correlation analyses controlling for BMI and sex revealed significant associations between behavioral factors, subjective stress, and physiological markers. Specifically, uncontrolled eating (UE) was correlated with %Δ QTc (r = 0.36, *p* < 0.05), with %Δ HR (r = 0.35, *p* < 0.05), and positively associated with fatty food desire (r = 0.35, *p* < 0.05). emotional eating (EE) was negatively related to healthy food desire (r = − 0.36, *p* < 0.05).

PSS was positively correlated with VAS stress (r = 0.35, *p* < 0.05), while VAS stress ratings were positively correlated with fatty food desire (r = 0.54, *p* < 0.01). Higher stress was also linked with increased anxiety toward fatty foods (r = 0.65, *p* < 0.001), with sweet food anxiety (r = 0.59, *p* < 0.001), and with healthy food anxiety (r = 0.57, *p* < 0.001). VAS anxiety was positively correlated with PSS (r = 0.35, *p* <0.05), with fatty food desire (r = 0.49, *p* < 0.01), with fatty food anxiety (r = 0.76, *p* < 0.001), with sweet food anxiety (r = 0.71, *p* < 0.001), and with healthy food anxiety (r = 0.67, *p* < 0.001). VR supermarket immersion was negatively correlated with EE (r = 0.51, *p* < 0.01), while VR stress was positively correlated with fatty food desire (r = 0.36, *p* < 0.05), with fatty food anxiety (r = 0.46, *p* < 0.01), with sweet food anxiety (r = 0.48, *p* < 0.01), and with healthy food anxiety (r = 0.42, *p* < 0.01) (Table 3 and Table 4).

Simple linear regression analyses further demonstrated that UE was significantly predicted by physiological and craving-related variables. The regression model with PSS %Δ QTc Fridericia was statistically significant [F(1,36) = 5.64, *p* = 0.02, R^2^ = 0.13, adjusted R^2^ = 0.11], as was the model with %Δ HR [F(1,36) = 6.03, *p* = 0.01, R^2^ = 0.14, adjusted R^2^ = 0.12]. Similarly, fatty food desire significantly predicted UE [F(1,36) = 6.07, *p* = 0.01, R^2^ = 0.14, Adjusted R^2^ = 0.12] (Table 5).

Mauchly’s test indicated that the assumption of sphericity was not violated for the within-subjects factor food (W = 0.96, χ^2^(2) = 1.26, *p* = 0.53). Therefore, results are reported with sphericity assumed. The repeated-measures ANOVA revealed a significant main effect of food type on craving, F(2,68) = 29.82, *p* < 0.001, η^2^ = 0.46, indicating that food cravings differed depending on food category. Multivariate tests confirmed this effect (Pillai’s Trace = 0.64, F(2,33) = 30.40, *p* < 0.001, η^2^ = 0.64). Post hoc pairwise comparisons (Bonferroni corrected) showed that cravings for sweet foods (mean = 4.86, SE = 0.26) were significantly higher than for fatty foods (mean = 3.91, SE = 0.22, *p* = 0.002) and healthy foods (mean = 3.05, SE = 0.22, *p* < 0.001). Regarding covariates, uncontrolled eating (UE) significantly moderated the effect of food type, F(2,68) = 5.00, *p* = 0.009, η^2^ = 0.12, suggesting that participants with higher UE scores reported stronger cravings for sweet and fatty foods. Neither cognitive restraint (CR), F(2,68) = 0.03, *p* = 0.97, η^2^ = 0.001, nor emotional eating (EE), F(2,68) = 1.37, *p* = 0.26, η^2^ = 0.03, significantly interacted with food type. Between-subjects effects were non-significant for UE (F(1,34) = 0.30, *p* = 0.58), CR (F(1,34) = 0.42, *p* = 0.52), and EE (F(1,34) = 0.78, *p* = 0.38), indicating that these factors did not independently predict overall craving intensity (Table 6 and Table 7).

A 2 × 3 mixed repeated measures ANOVA examined the effect of food type (sweet, fatty, healthy) and uncontrolled eating (UE group: low vs. high) on stress-induced food craving (VAS appetite) after VR-TSST exposure. There was a significant main effect of Food Type (F(1.93,435.41) = 168.98, *p* < 0.001, η^2^p = 0.43), indicating that cravings differed across food categories, with sweet foods rated highest. The food type × UE interaction was significant (F(1.93,435.41) = 16.49, *p* < 0.001, η^2^p = 0.07), showing that adolescents with high UE reported greater cravings for sweet and fatty foods than those with low UE. The main effect of UE group was not significant (F(1,226) = 0.05, *p* = 0.826), indicating similar overall craving levels between groups but distinct patterns across food types. These findings confirm that acute stress selectively increases cravings for palatable foods and that this effect is amplified among individuals with higher uncontrolled eating (Figure 4).

Hierarchical regression analyses were performed to examine whether there is a link between VR immersion, perceived stress, and eating desire after stress, including uncontrolled eating and food craving during supermarket exposure. In each model, perceived stress was Step 1, VR immersion at Step 2, and the interaction term (stress and immersion) at Step 3. None of the models were significant for uncontrolled eating (UE) In the supermarket immersion condition, the model including stress, immersion, and their interaction was not significant, F(3, 34) = 0.96, *p* = 0.42, R^2^ = 0.08, with a similar result observed in the classroom immersion condition, F(3,34) = 0.36, *p* = 0.79, R^2^ = 0.03. For general food desire and sweet desire in the classroom environment, the regression models were not significant, F(3,34) = 0.39, *p* = 0.75, R^2^ = 0.03, and F(3,34) = 0.86, *p* = 0.46, R^2^ = 0.07, respectively. In contrast, the model predicting fatty food desire in the virtual supermarket was significant, F(3,34) = 5.28, *p* < 0.001, R^2^ = 0.31 For sweet food desire, the model did not reach significance, F(3,34) = 1.10, *p* = 0.29, R^2^ = 0.08, and for healthy food desire, the regression model was also nonsignificant, F(3,34) = 1.37, *p* = 0.32, R^2^ = 0.10 (Table 8, Table 9 and Table 10).

## 4. Discussions

### 4.1. Relationship Between Eating Traits (UE, EE, CR) and Post-Stress Cravings

The present study demonstrates that acute stress exposure, elicited through the virtual Trier Social Stress Test (VR-TSST), significantly influenced adolescents’ subsequent food-related motivation. Specifically, cravings assessed immediately after the stress task during a virtual supermarket scenario were not uniformly elevated, but varied according to food type and baseline eating traits. The significant main effect of food category indicated that stress selectively increased the desire for high-reward foods (sweet and fatty), while interest in healthy foods remained comparatively low. These findings are consistent with previous research showing that stress induces a shift toward the consumption of energy-dense, palatable food. This reflects a “stress-eating” pattern driven by reward and emotional regulation mechanisms.

Partial correlations revealed that uncontrolled eating (UE) was positively associated with post-stress cravings for fatty and sweet foods. Emotional eating (EE) was negatively related to the desire for healthy foods. These findings are consistent with Debeuf et al. [28], who reported that daily stress is linked to stronger eating motives and cravings in adolescents. In their study of 109 participants (10–17 years), stress was consistently associated with increased desire to eat and hunger-driven eating across the week, and emotional eating had only a marginal moderating role. Maladaptive emotion regulation strategies did not significantly influence the relationship [28]. These results emphasize the dynamic nature of stress-related eating in youth and the need for further exploration of emotion regulation processes.

In line with this, Hyldelund et al. highlighted that stress selectively enhances reward-driven responses for high-fat, high-sugar foods. Using a randomized crossover design, they found that craving increased under stress, even though explicit liking and preference did not change. More than half of the participants changed their snack preferences after stress exposure, with only a few abstaining entirely. These results, together with previous findings, indicate that acute psychosocial stress can either increase the desire for palatable foods or reduce appetite. This reflects substantial individual variability in stress-related eating patterns [29].

Repeated-measures ANOVA revealed that post-stress cravings for sweet foods were significantly higher than those for fatty or healthy foods, particularly among participants with high UE scores. Specifically, participants with higher UE scores reported stronger cravings for both sweet and fatty foods. These findings mirror prior studies showing that high-calorie foods are associated with stronger cravings than low-calorie foods, in both real and virtual contexts [30,31].

Reents and Pedersen [30] investigated how food type and social context in VR environments affect food cravings. In their study, 87 female students were exposed to four VR scenarios involving either low- or high-calorie foods in kitchen or restaurant settings. The results showed significantly higher cravings after exposure to high-calorie foods compared to low-calorie ones, regardless of the social context. Interestingly, social setting (being alone vs. with friends) did not significantly influence craving, and only limited associations emerged between BMI, eating disorder symptoms, and craving responses. Overall, this study supports the idea that VR-based cue exposure reliably tends to increase food cravings, particularly for high-calorie foods, and can be used as a research and intervention tool.

Similarly, Ferrer-Garcia et al. [31] explored how food type and social context in VR environments shape cravings in a non-clinical female sample. Eighty-seven students were immersed in four VR scenarios featuring either low- or high-calorie foods in kitchen or restaurant settings. Findings revealed that exposure to high-calorie foods consistently triggered stronger cravings than low-calorie foods, while social context had no significant effect. Although BMI and eating disorder symptoms showed no overall associations with craving, a negative correlation emerged between BMI and craving in the low-calorie restaurant scenario. The study concluded that VR-based cue exposure is a reliable method for eliciting cravings and may have therapeutic applications.

In our study, neither cognitive restraint (CR) nor emotional eating (EE) was significantly associated with variations in craving intensity, which contrasts with some previous research. This discrepancy may be due to sample characteristics or the immersive VR setting. Notably, Werthmann et al. [32] demonstrated that healthy food imagery can increase cravings for healthier options, underlining the role of context in shaping motivation. In their experiment, participants engaged in guided imagery tasks of consuming healthy foods, which significantly increased both cravings for healthy foods and motivation to adopt healthier eating. This suggests that while stress tends to enhance hedonic drives for high-calorie foods, contextual factors such as guided imagery can shift motivation toward healthier choices.

Importantly, the significant food type × uncontrolled eating interaction revealed that individuals with higher baseline uncontrolled eating (UE) scores, as measured by the TFEQ, exhibited greater stress-induced cravings for sweet and fatty foods. This effect was observed when comparing them with their low-UE peers. This suggests that pre-existing tendencies toward disinhibited eating amplify the hedonic response to stress, consistent with models of individual susceptibility to stress-related overeating. The absence of a main effect of UE group on overall craving levels implies that the stress response does not uniformly elevate craving intensity. Instead, it modulates the craving profile, enhancing the appeal of certain food types in vulnerable individuals.

Together, these results highlight that the interplay between acute stress and baseline eating traits determines stress-related food motivation. This has important implications for the prevention of stress-induced overeating and weight gain in adolescents. The combined use of immersive virtual stress exposure and a virtual supermarket paradigm offers a novel, ecologically valid approach for exploring real-world eating behavior under controlled laboratory conditions.

### 4.2. Relationship Between Physiological Changes and Post-Stress Cravings

Linear regression models indicated that alterations in QTc, HR, and fatty food desire were significantly linked to UE. This suggests that physiological stress responses may directly shape eating behaviors by amplifying hedonic drives. This is consistent with evidence that HPA activation and glucocorticoid release stimulate appetite for palatable foods [32]. Our results highlight the role of autonomic and neuroendocrine reactivity as mediators between psychological stress and maladaptive food choice. Supporting this interpretation, Hill et al. [33] reported in a meta-analysis that stress-related unhealthy eating behaviors emerge as early as 8–9 years of age, with adolescents showing greater consumption of energy-dense foods and reduced intake of healthier options. Additional evidence from Caso et al. [34] showed that academic stress predicted higher unhealthy food intake in Italian undergraduates, whereas French students reported reduced junk food intake under stress. Moderation by EE and BMI was observed for sweet intake and snacking, but no significant role was found for restrained eating. Building on this evidence, our findings extend prior research by linking stress-related physiological markers (QTc and HR) to eating tendencies, bridging biological reactivity and behavioral outcomes.

ECG modifications were reported by Padfield et al. [35], where adolescent females with anorexia nervosa exhibited significantly longer QTc intervals at peak exercise compared with healthy controls (442 ± 29 ms vs. 422 ± 19 ms). This difference indicates a reduced repolarization reserve under sympathetic activation. Martín Rivada et al. [36] observed that PR segment changes and alterations in the QRS complex were more frequent in adolescents with anorexia nervosa. Significant differences in the prevalence of prolonged QTc between cases and controls were not found. Two patients exhibited QT interval changes. These findings support the interpretation that ECG markers, particularly those reflecting conduction and repolarization dynamics, may serve as sensitive indicators of altered autonomic and myocardial function in underweight adolescents [36].

Tomar et al. [37] investigated 90 individuals stratified by Southeast Asian BMI criteria (normal, overweight, obese) to examine ventricular repolarization parameters and HRV. Their results demonstrated that obese participants had significantly longer QTc and Tpeak–Tend (Tpe) intervals compared to normal-weight subjects. QTc interval was correlated with all HRV parameters, reflecting the influence of autonomic alterations. In contrast, Tpe changes were independent of HRV, suggesting direct effects of obesity on cardiac or visceral fat. The study concluded that electrocardiographic changes in obesity are linked both to autonomic imbalance and structural cardiac changes, highlighting the potential role of ECG as a screening tool for cardiovascular risk in obese individuals [37].

Benayon et al. [38] research has reported QTc interval prolongation among individuals with eating disorders, often attributed to altered autonomic and metabolic regulation. Early studies suggested that QTc abnormalities were inherent to anorexia nervosa. More recent findings indicate that these changes are primarily influenced by extrinsic factors such as electrolyte imbalance and psychotropic medication use. This supports the view that QTc variability reflects underlying physiological dysregulation associated with disordered eating patterns, which may also emerge in non-clinical populations under stress exposure [38].

In line with these findings, our results suggest that acute psychosocial stress may transiently prolong QTc intervals through increased sympathetic activation and reduced vagal tone. These mechanisms have also been implicated in stress-related cravings for high-calorie foods. This autonomic imbalance may thus serve as a physiological correlate of heightened appetitive motivation under stress exposure in adolescents.

The use of portable ECG devices, such as WIWE, is supported by numerous validation studies showing that handheld and mobile technologies can reliably capture key cardiac parameters. Ahmadi-Renani et al. [39] demonstrated that a handheld single-lead ECG device achieved a 98% compatibility rate with a standard 12-lead ECG across 300 patients, with no critical abnormalities missed and high accuracy in detecting pathological intervals. Similarly, Bovenkerk et al. [40] demonstrated that a 6-lead mobile ECG provides reliable PR, QRS, and QT measurements compared with standard 12-lead ECGs. This supports its use for remote cardiac monitoring. In healthy adults, Klier et al. [41] reported strong concordance between single-lead recordings obtained with portable devices (KardiaMobile and Apple Watch 4) and 12-lead ECGs, despite a tendency to slightly underestimate QT and QTc values. Taken together, these findings highlight that portable ECG systems are not only accurate and safe for screening and rhythm detection but also cost-effective and convenient for large-scale or remote monitoring. Within this context, the use of WIWE in our study provides a valid and reliable means of assessing acute stress-induced changes in QTc, HR, and PQ intervals in adolescents [41].

### 4.3. Virtual TSST and Food Exposure

Our findings showed that perceived stress during the VR-TSST was positively associated with physiological markers, such as QTc changes, and with increased desire for high-calorie foods, including sweets and fatty products. Immersion was linked to greater anxiety toward sweet foods, pointing to a more complex interaction between engagement in VR and food-related cognitions. Among all regression models, only the one examining fatty food desire reached statistical significance (F(3,34) = 5.28, *p* < 0.001, R^2^ = 0.31), indicating that stress, VR immersion, and their interaction were associated with stronger cravings for high-fat foods after stress exposure. This suggests that the immersive virtual supermarket effectively reflected real-life stress-related eating patterns. In contrast, no significant associations were observed for uncontrolled eating, general, sweet, or healthy food desire, suggesting that stress is selectively related to motivation for high-calorie, hedonic foods. These findings are consistent with prior evidence that acute psychosocial stress can promote preference for energy-dense foods, particularly in adolescents, and extend this knowledge into a VR-based environment. The same virtual classroom environment has been applied in previous research by Moïse-Richard A. et al. [42], with school-age children and adolescents who stutter. The objective of that study was to evaluate whether a virtual classroom could serve as a relevant and feasible tool for exposing participants to feared public speaking situations. The researchers compared self-reported anxiety levels and stuttering severity in front of a virtual class with those observed in a real audience setting. Their results showed that although anticipatory anxiety was higher when speaking in front of a real audience. Both anxiety levels and stuttering severity in the virtual classroom were comparable to real-life conditions and significantly greater than in a neutral virtual setting (empty virtual apartment). These findings confirm that the virtual classroom paradigm used in the present study has been previously validated as a reliable and ecologically valid tool for inducing social-evaluative stress, supporting its use in virtual adaptations of the Trier Social Stress Test (TSST-VR) [42].

Van der Waal et al. [43] reported that food cue responses in VR approximate those in real life. The study investigated whether food cue responses in virtual reality (VR) resemble those in real-life situations and how hunger affects these responses. Using a within-subjects design (N = 54), participants were exposed to food and non-food cues in both VR and real-life settings under hungry and satiated conditions. Exposure to food increased cravings in both VR and real life, though cravings were weaker in VR. It suggests that VR effectively elicits craving responses similar to real life, making it a promising tool for measuring food-related behavior and for interventions like cue exposure therapy. Using salivation as a physiological marker, it was found that exposure to food cues induced greater salivation than exposure to non-food cues. This effect was significantly weaker in VR compared to real settings. Moreover, the hunger state did not significantly modulate these responses. While VR food cues can trigger craving-related physiological reactions, the magnitude remains lower compared to real-life exposure. It suggests that virtual environments are effective models, but may not fully replicate the intensity of real-world food cue responses.

Ecker et al. [10] demonstrated that VR-based Trier Social Stress Tests reliably induce stress responses, with a lower impact than in vivo protocols. In the study, a virtual version of the Trier Social Stress Test for Children (TSST-C) was compared with the traditional in-person protocol in adolescents. Both versions successfully elicited multimodal stress responses across the hypothalamic–pituitary–adrenal (HPA) axis and autonomic nervous system, as indicated by changes in cortisol, alpha-amylase, heart rate, and heart rate variability. Although the virtual condition produced lower cortisol responses than the real TSST-C, it still induced significant physiological and subjective stress reactions, confirming the validity of the virtual paradigm for studying stress reactivity. This supports the utility of VR as a controlled, replicable tool for studying stress-related eating behaviors. However, limitations must be acknowledged, including the modest sample size, the short exposure duration, and the reliance on self-reported craving measures rather than actual food intake.

Virtual reality (VR) enables users to engage in realistic, immersive environments, allowing simulated experiences and behaviors to closely mirror those occurring in real life. Presence reflects the human tendency to respond to virtually generated sensory inputs as if they were authentic. When designing VR scenarios to study eating behavior, participants need to have a high sense of presence while immersed.

### 4.4. Limitations

A limitation of this study is the relatively small sample size, which reduces the statistical power of the analyses and restricts the generalizability of the findings to the wider adolescent population. Another consideration is the use of a portable ECG device (WIWE). Although this system has been validated in previous studies, it presents certain constraints compared to standard clinical-grade equipment, such as potential susceptibility to motion artifacts and lower resolution in capturing specific parameters. In addition, the menstrual cycle stage was not controlled in female participants, which may have influenced physiological and craving responses. Food craving was assessed through self-report ratings, limiting the ability to capture actual consumption behavior. The VR supermarket environment may have introduced novelty or gaming effects that influenced food-related decisions independently of stress mechanisms. Due to the limited sample size, realism scores were not included as covariates or in partial correlation analyses to avoid model overfitting. Given the small sample size, the regression results should be viewed as exploratory, since limited data may lead to unstable parameter estimates. Future research should include larger and more diverse samples, control for hormonal variations, examine whether perceived realism moderates stress-related food choice and craving responses, and combine subjective and behavioral measures to provide a more comprehensive understanding of stress-related eating patterns.

## 5. Conclusions

In conclusion, this study demonstrates that acute stress exposure through a virtual Trier Social Stress Test (VR-TSST) produces measurable changes in both physiological markers and eating-related motivation among adolescents. The virtual reality supermarket paradigm revealed that post-stress cravings were not generalized but selectively increased for sweet and fatty foods, particularly in individuals with higher uncontrolled eating scores. These findings indicate that stress interacts with baseline eating traits to shape food-related motivation, offering new insights into stress-induced appetitive responses during adolescence.

Beyond its theoretical contributions, the study introduces a novel, ecologically valid experimental framework that integrates immersive VR stress exposure and food-choice simulation to investigate psychophysiological and behavioral responses under controlled yet realistic conditions. This approach may contribute to the development of personalized prevention and intervention strategies targeting stress-related overeating by identifying individuals most susceptible to reward-driven eating under stress.

## Figures and Tables

**Figure 1 nutrients-17-03924-f001:**
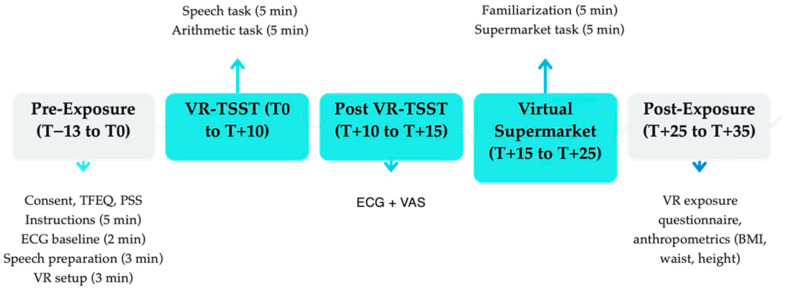
Study protocol. Schematic summary of the procedure.

**Figure 2 nutrients-17-03924-f002:**
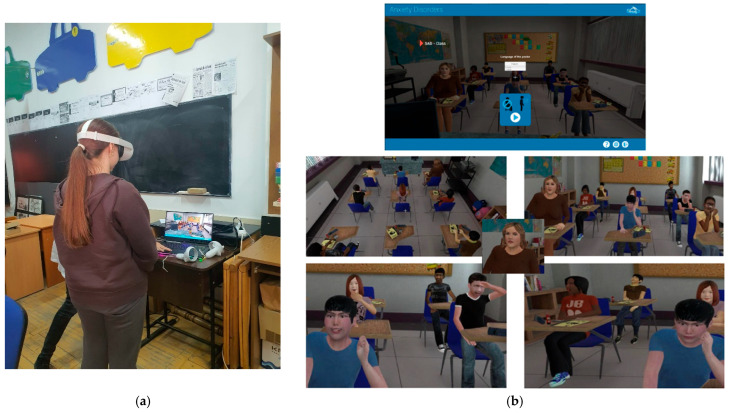
Screenshot from the VR applications. (**a**) Participant explores the VR classroom. (**b**) Screenshot from the VR classroom.

**Figure 3 nutrients-17-03924-f003:**
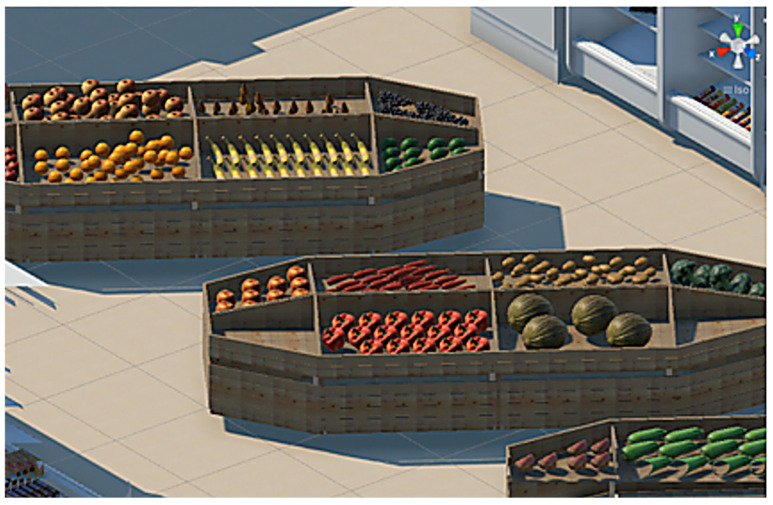
Screenshot from the VR supermarket.

**Figure 4 nutrients-17-03924-f004:**
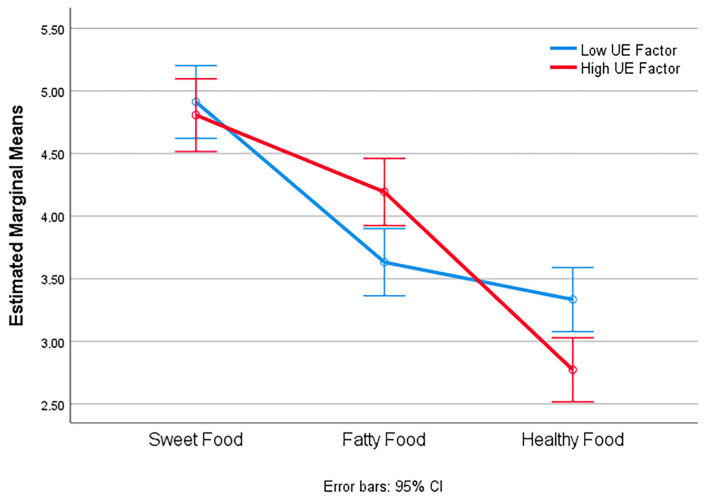
Estimated marginal means of food craving (VAS appetite) for sweet, fatty, and healthy foods during the virtual reality supermarket task, following exposure to the VR Trier Social Stress Test (VR-TSST). The figure illustrates the interaction between food type (sweet, fatty, healthy) and uncontrolled eating (UE) group (low vs. high) on food cravings measured with VAS during the virtual reality supermarket task, performed after exposure to the VR Trier Social Stress Test (VR-TSST). Adolescents with high UE reported higher cravings for sweet and fatty foods compared with those with low UE, while craving for healthy foods decreased in both groups.

**Table 1 nutrients-17-03924-t001:** Pre-exposure characteristics of the studied participants.

Characteristics	Participants (N = 38)	Sex (%)	
		Boys	Girls
Age (years)	15.82 ± 0.12	-	-
Anthropometric parameters			-
Body mass index (BMI) (kg/m^2^)	20.01 ± 0.67		
Underweight (<−1 SD) (%, N) ^1^	42.1 (16)	25	75
Normal weight (−1 SD–+1 SD) (%, N) ^1^	36.8 (14)	28.6	71.4
Overweight (>+1 SD) (%, N) ^1^	21.1 (8)	50	50

^1^ According to age-specific BMI cutoff points published by the WHO. Continuous variables were presented as mean (SD). Categorical variables are presented as frequencies and percentages.

**Table 2 nutrients-17-03924-t002:** Descriptive statistics of ECG, stress and anxiety assessment, and eating behavior as a response to the VR exposure experiment.

	VR TSST				
Parameter	Pre VR-TSST	Post VR-TSST	*p*-Value ^1^	Cohen’s d (95% CI) ^3^	Cronbach’s α
Primary outcomes					
Electrocardiogram measures					
QTc Fridericia (ms)	431.73 ± 34.55	429.92 ± 37.90	0.48	0.05 (−3.44–7.06)	-
PQ (ms)	146.13 ± 18.08	140.58 ± 17.42	0.02	0.31 (0.76–10.34)	-
HR (bpm)	89.32 ± 13.29	89.45 ± 14.47	0.85	−0.01 (−1.55–1.29)	-
Secondary outcomes					
Perceived stress	Questionnaires			-	-
Perceived Stress Scale (PSS)	33.18				
Stress and Anxiety Assessment (VAS method)					
How stressed are you feeling right now?	-	3.47 ± 1.91	-	-	-
How anxious are you feeling right now?	-	3.32 ± 1.84	-	-	-
Eating behavior parameters ^2^					
CR-factor	20.21 ± 5.13	-	-	-	0.80
UE-factor	13.89 ± 3.79	-	-	-	0.76
EE-factor	12.08 ± 3.96	-	-	-	0.73
Appetite and craving ratings (VAS method)					
How strong is your desire to eat right now?	-	3.84 ± 2.21	-	-	-
How strong is your desire to eat sweets right now?	-	2.84 ± 1.65	-	-	-
	Supermarket exposure				
	VR Supermarket		Post VR Supermarket		
Primary Outcomes					
Fatty food desire	3.91 ± 1.49		-		
Fatty food anxiety	2.56 ± 1.61				
Sweet food desire	4.85 ± 1.58		-		
Sweet food anxiety	2.57 ± 1.41		-		
Healthy food desire	3.0.5 ± 1.42		-		
Healthy food anxiety	2.10 ± 1.23		-		
	Perceived VR exposure				
Secondary Outcomes			-		
VR supermarket immersion	4.55 ± 1.85				
VR classroom immersion	4.79 ± 1.66		-		
VR supermarket realism	3.32 ± 1.64		-		
VR classroom realism	3.13 ± 1.54		-		
VR stress	3.82 ± 1.72		-		
VR discomfort	1.84 ± 1.17		-		
VR performance	4.87 ± 1.39		-		

^1^ The paired *t*-test was used to compare two related samples. A *p*-value < 0.05 was considered statistically significant. ^2^ Measured with the Three-Factor Eating Questionnaire. Continuous variables were presented as mean (SD). ^3^ Size effects were reported with Cohen’s d and 95% confidence intervals (CI) Note: HR = heart rate; QTc = corrected QT interval; PQ = PQ interval; TFEQ (Three-Factor Eating Questionnaire) scales (CR = cognitive restraint; UE = uncontrolled eating; EE = emotional eating); VAS = Visual Analog Scale; VR = virtual reality.

**Table 3 nutrients-17-03924-t003:** Partial correlation with covariables (BMI and sex) in 38 adolescents.

	1	2	3	4	5	6	7	8	9	10	11
Factor UE		−0.27	0.60 ***	0.31	0.36 *	−0.14	0.357 *	0.02	−0.02	0.04	−0.19
Factor CR			0.04	0.16	−0.17	−0.12	−0.28	−0.01	−0.01	−0.04	0.07
Factor EE				0.17	0.30	−0.30	0.32	−0.22	−0.23	−0.05	−0.17
PSS score					0.11	−0.19	0.23	0.25	0.35 *	0.29	0.12
%Δ QTc						−0.10	0.76 ***	−0.04	0.11	0.00	0.04
%Δ PQ							−0.12	0.01	0.06	−0.11	0.22
%Δ HR								0.07	0.21	−0.13	−0.20
VAS stress									0.83 ***	0.22	0.08
VAS anxiety										0.17	0.02
VAS appetite											0.29
VAS sweet craving											

Note: UE = uncontrolled eating; CR = cognitive restraint; EE = emotional eating; PSS = Perceived Stress Scale; %Δ = percent change; QTc = corrected QT interval; PQ = PQ interval; HR = heart rate; VAS = Visual Analog Scale; * *p* < 0.05, *** *p* < 0.001.

**Table 4 nutrients-17-03924-t004:** Partial correlation with covariables (BMI and sex) in 38 adolescents.

	1	2	3	4	5	6	7	8	9	10	11
Fatty food desire	0.35 *	−0.30	0.025	0.22	0.01	0.31	0.16	0.53 ***	0.49 **	0.15	0.16
Sweet food desire	−0.04	−0.03	−0.00	0.23	0.06	0.17	0.10	0.15	0.24	0.33 *	0.39 *
Healthy food desire	−0.22	−0.15	−0.36 *	−0.10	−0.16	0.41 *	−0.05	0.12	0.12	0.15	0.18
Fatty food anxiety	−0.03	0.00	−0.10	0.14	0.02	0.13	0.18	0.65 ***	0.76 ***	0.01	−0.01
Sweet food anxiety	−0.06	0.02	−0.03	0.13	0.11	0.08	0.30	0.59 ***	0.71 ***	−0.03	0.00
Healthy food anxiety	−0.03	−0.01	−0.08	0.10	0.13	0.00	0.30	0.57 ***	0.67 ***	−0.04	−0.01
VR supermarket immersion	−0.12	0.33	−0.51 ***	0.04	−0.11	0.07	0.03	0.25	0.25	0.26	−0.01
VR supermarket realism	0.15	−0.06	−0.05	0.19	−0.25	0.13	0.01	−0.06	−0.06	0.09	−0.01
VR supermarket stress	−0.05	0.12	0.02	0.02	0.00	0.11	0.17	0.27	0.36 *	−0.23	−0.25

* *p* < 0.05, ** *p* < 0.01, *** *p* < 0.001.

**Table 5 nutrients-17-03924-t005:** Linear regression for UE predictor of change in eating behavior and heart rate parameters.

	Unstd. Beta	Std. Error	Std Beta	t	Sign. *p*
Factor CR	−0.40	0.21	−0.29	−1.87	0.07
Factor EE	0.67	0.18	0.52	3.66	0.00
PSS	0.12	0.06	0.28	1.76	0.08
%Δ QTc Fridericia	0.51	0.21	0.36	2.37	0.02
%Δ PQ	−0.06	0.09	−0.11	−0.70	0.48
%Δ HR	0.40	0.16	0.37	2.45	0.01
VAS stress	0.10	0.44	0.39	0.23	0.81
VAS anxiety	−0.02	0.46	−0.01	−0.06	0.95
VAS sweet craving	−0.44	0.51	−0.14	−0.86	0.39
VAS appetite	0.22	0.38	0.09	0.57	0.56
Fatty food desire	1.30	0.53	0.38	2.46	0.01
Sweet food desire	−0.12	0.53	−0.03	−0.22	0.82
Healthy food desire	−0.74	0.58	−0.20	−1.27	0.21
Fatty food anxiety	−0.10	0.52	−0.03	−0.19	0.84
Sweet food anxiety	−0.09	0.60	−0.02	−0.16	0.87
Healthy food anxiety	−0.00	0.76	−0.00	−0.00	0.99

Note: CR = cognitive restraint; EE = emotional eating; PSS = Perceived Stress Scale; %Δ = percent change; QTc = corrected QT interval; PQ = PQ interval; HR = heart rate; VAS = Visual Analog Scale.

**Table 6 nutrients-17-03924-t006:** ANOVA tests of within-subject effects.

		*df*	F	Sig.	Partial Eta Squared
Food	Sphericity assumed	2	29.82	0.00	0.46
Food ZUE_UNCONTROLLED_EATING	Sphericity assumed	2	5.00	0.00	0.12
Food ZCR_COGNITIVE_RESTRAINT	Sphericity assumed	2	0.02	0.97	0.00
Food ZEE_EMOTIONAL_EATING	Sphericity assumed	2	1.37	0.26	0.03

**Table 7 nutrients-17-03924-t007:** Pairwise comparisons.

(I) Food	(J) Food	Std. Error	Sig. b	95% Confidence Interval for Difference	
				Lower Bound	Upper Bound
1. Fatty food desire	2. Sweet food desire	0.25	0.00	−1.58	−0.30
	3. Healthy food desire	0.21	0.00	0.31	1.40
2. Sweet food desire	1. Fatty food desire	0.25	0.00	0.30	1.58
	3. Healthy food desire	0.23	0.00	1.22	2.39
3. Healthy food desire	1. Fatty food desire	0.21	0.00	−1.40	−0.31
	2. Sweet food desire	0.23	0.00	−2.39	−1.22

**Table 8 nutrients-17-03924-t008:** Hierarchical regression predicting uncontrolled eating from stress, VR immersion, and their interaction.

Model	Sum of Squares	Mean Square	F	Sig.	R	R Square	Adjusted R Square
**UE and supermarket immersion**							
1. Stress	27.68	13.84	0.51	0.16	0.02	−0.02	−0.02
2. Supermarket immersion	75.81	25.27	0.95	0.27	0.07	−0.00	−0.00
3. Stress and supermarket interaction	27.25	27.25	1.03	0.16	0.02	0.00	0.00
**UE and classroom immersion**							
1. Stress	1.49	1.49	0.05	0.81	0.03	0.00	−0.02
2. Classroom immersion	10.06	5.03	0.18	0.83	0.10	0.01	−0.04
3. Stress and classroom interaction	29.64	9.88	0.35	0.78	0.17	0.03	−0.05

**Table 9 nutrients-17-03924-t009:** Hierarchical regression predicting food desire/sweets desire, stress, VR immersion, and their interaction.

Model	Sum of Squares	Mean Square	F	Sig.	R	R Square	Adjusted R Square
Food desire and classroom immersion							
1. Stress	4.92	4.92	1.00	0.32	0.16	0.02	0.00
2. Classroom immersion	5.39	2.69	0.53	0.58	0.17	0.03	−0.02
3. Stress and classroom interaction	6.14	2.04	0.39	0.75	0.18	0.03	−0.05
Sweets desire and classroom immersion							
1. Stress	1.41	1.41	0.51	0.47	0.11	0.01	−0.01
2. Classroom immersion	6.84	3.42	1.27	0.29	0.26	0.06	0.01
3. Stress and classroom interaction	7.15	2.38	0.86	0.46	0.26	0.07	−0.01

**Table 10 nutrients-17-03924-t010:** Hierarchical regression predicting fatty food/sweets/healthy food desire, and stress, VR immersion, and their interaction.

Model	Sum of Squares	Mean Square	F	Sig.	R	R Square	Adjusted R Square
**Fatty food desire and supermarket immersion**							
1. Stress	25.04	25.04	15.66	0.00	0.55	0.30	0.28
2. Supermarket immersion	25.44	12.72	7.79	0.00	0.55	0.30	0.26
3. Stress and supermarket interaction	26.27	8.75	5.28	0.00	0.56	0.31	0.25
**Sweet food desire and supermarket immersion**							
1. Stress	1.35	1.35	0.52	0.47	0.12	0.01	−0.01
2. Supermarket immersion	7.96	3.98	1.63	0.21	0.29	0.08	0.03
3. Stress and supermarket interaction	8.26	2.75	1.10	0.36	0.29	0.08	0.00
**Healthy food desire and supermarket immersion**							
1. Stress	3.69	3.69	1.85	0.18	0.22	0.04	0.02
2. Supermarket immersion	8.08	4.04	2.10	0.13	0.32	0.10	0.05
3. Stress and supermarket interaction	8.16	2.72	1.37	0.26	0.32	0.10	0.03

## Data Availability

The data presented in this study are available on request from the corresponding author. The data presented in this study are not publicly available due to privacy and ethical restrictions related to minors’ data.

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
