# Peer review of "Virtual Reality Trier Social Stress and Virtual Supermarket Exposure: Electrocardiogram Correlates of Food Craving and Eating Traits in Adolescents"

_nutrients, 2025, doi:10.3390/nu17243924_

Round 1
Reviewer 1 Report
Comments and Suggestions for Authors
Overall this paper outlines some novel testing of stress on grocery shopping in adolescents. The paper could use some significant revisions in order to be appropriate for publication. Overall I found the results somewhat difficult to read, as the analyses were dense and the tables were also quite dense. This made interpreting the discussion somewhat difficult.
Major comments
The introduction is not appropriately referenced. There are not references for each factual statement, as well as the current references do not seem to match the statements. For example, reference 1, the reference describes a study examining psychosocial well being with impulsiveness on sweet and fat propensity, but the sentence the reference appears in describes heightened brain reward activation, reinforcement, and motivation, which is not discussed in the referred paper. The authors should first carefully ensure currently referenced articles match the statement and then carefully review the paper for statements that require references. I had a difficult time assessing the paper without appropriate references for statements, as it was unclear if statements were author hypotheses versus previously researched hypotheses. The methodology reports some use of AI to rephrase/language editing. I think the authors should very carefully review if the LLM included references the authors did not mean to include, as they do not always seem to match the statements.
It is not clear what the purpose of the grocery shopping test was. What were adolescents asked to do? There is a note that they had to “find” nine items in three categories. What was the goal of this task? Based on the introduction I assumed it was a choice task, but that was not clear in the methods. The interest of this paper hinges on the outcomes associated with this task and I was not able to assess it appropriately.
Other comments
Introduction
- Need a reference for link between dietary quality and obesity prevalence
- Need references for all of paragraph 2
- Paragraph 2 is a mess
- Paragraph 3 – again, no references for the points
- Stress has different effects on food consumption/energy intake – this is not a balanced review in the introduction
- First sentence in paragraph 3 – this sentence is ambiguous. What population is this statement discussion, 35% and 60% of who? The reference is a summary of a symposium, so its not clear where this information is coming from. The last sentence also discussed decreasing calorie intake by preferring high calorie foods - was this a typo?
- This is not a study of eating behavior – its food choices, there is no eating involved
Methods
- It’s not clear if the virtual TSST has been validated
- Information on approval of study procedures?
- A lot of acronyms without definitions (e.g QRS, QT, WIWE device). Please assume that your readers will need initial spelling out of each abbreviation/acronym used.
- This should also show up in all tables, e.g. there should be a table inset with the acronym definitions
- VR-TSST – who prompted the student if participants stopped before time ended – a virtual prompt or a prompt from the experimenter in the room?
- Were participants tested individually?
- I don’t understand the virtual supermarket task – in the procedures it mentions directing students to select certain items? Or were students allowed to choose any items they wanted? How was this related to real-world consequences, could they eat some of the items they chose, did they get items to take home? Was there a hypothetical scenario about the shopping that provided external validity to the task?
- Is there enough power to detect moderation/mediation effects with n = 38?
- Why not use hierarchical regression analysis to see if moderation effects significantly improved the model?
Author Response
Comment 1.
Major comments
The introduction is not appropriately referenced. There are not references for each factual statement, as well as the current references do not seem to match the statements. For example, reference 1, the reference describes a study examining psychosocial well being with impulsiveness on sweet and fat propensity, but the sentence the reference appears in describes heightened brain reward activation, reinforcement, and motivation, which is not discussed in the referred paper. The authors should first carefully ensure currently referenced articles match the statement and then carefully review the paper for statements that require references. I had a difficult time assessing the paper without appropriate references for statements, as it was unclear if statements were author hypotheses versus previously researched hypotheses.
Response:
Thank you for the helpful observation. The introduction has been completely revised to ensure that all factual statements are supported by accurate and relevant references. Each citation has been verified to match the corresponding statement and replaced or supplemented where necessary.
Comment
The methodology reports some use of AI to rephrase/language editing. I think the authors should very carefully review if the LLM included references the authors did not mean to include, as they do not always seem to match the statements.
Response:
We thank you for the helpful observation. The large language model (LLM, ChatGPT) was used exclusively for language editing and phrasing, not for generating content or inserting references. All references were carefully verified and cross-checked by the authors to ensure full consistency with the statements in the text.
Following the suggestion, the introduction was revised to ensure that every factual statement is appropriately supported by a valid source. For example, the following references were added and verified:
(Birch & Doub, 2014, as cited in Do et al., 2024) – to support the role of environmental and emotional factors in shaping eating behavior during childhood and adolescence.
Lines 127, 129-130
Comment
It is not clear what the purpose of the grocery shopping test was. What were adolescents asked to do? There is a note that they had to “find” nine items in three categories. What was the goal of this task? Based on the introduction I assumed it was a choice task, but that was not clear in the methods. The interest of this paper hinges on the outcomes associated with this task and I was not able to assess it appropriately.
Response:
Thank you for your constructive feedback. We have added in the Introduction sectiob lines 279-281
Lines 385-395 were added, page 6.
Comment
Other comments
Introduction
Need a reference for link between dietary quality and obesity prevalence
Need references for all of paragraph 2
Paragraph 2 is a mess
Paragraph 3 – again, no references for the points
Stress has different effects on food consumption/energy intake – this is not a balanced review in the introduction
First sentence in paragraph 3 – this sentence is ambiguous. What population is this statement discussion, 35% and 60% of whom? The reference is a summary of a symposium, so its not clear where this information is coming from. The last sentence also discussed decreasing calorie intake by preferring high calorie foods - was this a typo?
This is not a study of eating behavior – its food choices, there is no eating involved
Response
We thank you for the constructive feedback. The introduction was substantially revised to improve clarity, structure, and referencing. Each factual statement is now supported by appropriate citations from peer-reviewed sources. Paragraphs 2 and 3 were modified.
Comment
Methods
It’s not clear if the virtual TSST has been validated
Response:
Thank you for this helpful observation. We have clarified in the Discussion section that the virtual classroom environment used in this study has been previously validated as an effective and ecologically valid tool for inducing social-evaluative stress. - Moïse-Richard et al. (2021)
Lines 1102-1113
Comment
Information on approval of study procedures?
Response:
We have added lines 334-336 were added, page 5 ( 2.2.1 Preparatory phase)
Comment
A lot of acronyms without definitions (e.g QRS, QT, WIWE device). Please assume that your readers will need initial spelling out of each abbreviation/acronym used.
This should also show up in all tables, e.g. there should be a table inset with the acronym definitions
Response:
We have added in the text and Notes for table 2, lines 653-655; table 3, lines 657-659; table 5, lines 666-668. A new section was added on page 22, lines 1243-1249.
Comment
VR-TSST – who prompted the student if participants stopped before time ended – a virtual prompt or a prompt from the experimenter in the room?
Were participants tested individually?
I don’t understand the virtual supermarket task – in the procedures it mentions directing students to select certain items? Or were students allowed to choose any items they wanted? How was this related to real-world consequences, could they eat some of the items they chose, did they get items to take home? Was there a hypothetical scenario about the shopping that provided external validity to the task?
Response:
Lines 379 – 389 were revised and modified.
Comment
Is there enough power to detect moderation/mediation effects with n = 38?
Response:
A new section was added at page 10, lines 532-538.
Comment
Why not use hierarchical regression analysis to see if moderation effects significantly improved the model?
Abstract lines 109-114
Statistical Analysis lines 556-561
Section 3.2 -Lines 576-583
Table 8, 9,10
Discussion section 1092-1098
Reviewer 2 Report
Comments and Suggestions for Authors The manuscript presents an interesting and timely investigation into the relationship between stress, physiological responses, and food cravings in adolescents using virtual reality (TSST-VR) and ECG monitoring. The integration of immersive technology with psychophysiological measures is innovative and holds promise for understanding eating behavior in youth. Major Strengths:
- Novel use of VR-TSST and virtual supermarket to assess stress-induced food cravings.
- Combined subjective (VAS, TFEQ) and objective (ECG) measures.
- Appropriate statistical analyses with control variables (BMI, sex).
Areas for Improvement:
- Language and Clarity:
The English language requires improvement for clarity and academic tone. There are numerous grammatical issues, awkward phrasings, and typographical errors throughout the manuscript. A professional language editing service is recommended. - Sample Size and Power:
The sample size (N = 38) is relatively small, limiting the generalizability and statistical power of the findings. The authors should acknowledge this limitation more explicitly in the discussion and consider it when interpreting non-significant results. - Psychometric Validation:
While the TFEQ-R21C is mentioned, no internal consistency reliability (e.g., Cronbach’s alpha) is reported. Please include reliability statistics to support the validity of the questionnaire in this adolescent sample. - Moderator Analysis Underutilized:
VR immersion and realism scores were collected but not effectively integrated into the analysis. It would strengthen the paper to explore whether immersion moderates the relationship between stress and food craving. - Causal Language:
Please revise instances where correlational results are interpreted causally (e.g., “predicted” vs. “associated with”). This is important to avoid overinterpretation of cross-sectional or quasi-experimental data. - Figures and Visual Aids:
While tables are comprehensive, the manuscript would benefit from visual representations (e.g., ECG changes over time, interaction plots for UE and food type). Consider adding figures to enhance readability and impact. - Discussion Depth:
The discussion could be improved by comparing findings more thoroughly with existing VR-based eating behavior studies and addressing potential mechanisms (e.g., autonomic regulation, reward processing) underlying the observed effects.
Conclusion:
This is a valuable and original contribution to the field. With minor revisions, particularly in language clarity, methodological transparency, and interpretive caution, this manuscript would be suitable for publication.
Author Response
Comment 1. Language and Clarity:
The English language requires improvement for clarity and academic tone. There are numerous grammatical issues, awkward phrasings, and typographical errors throughout the manuscript. A professional language editing service is recommended.
Response
Thank you for your helpful observation. We carefully revised the entire manuscript to improve clarity, grammar, and academic tone. Several sentences were rephrased to ensure fluency and precision, and typographical errors were corrected throughout the text. Language quality has been substantially improved in the revised version.
Comment 2. Sample Size and Power:
The sample size (N = 38) is relatively small, limiting the generalizability and statistical power of the findings. The authors should acknowledge this limitation more explicitly in the discussion and consider it when interpreting non-significant results.
Response:
Information was added in the limitations sections, lines 1990-1193
Comment 3. Psychometric Validation:
While the TFEQ-R21C is mentioned, no internal consistency reliability (e.g., Cronbach’s alpha) is reported. Please include reliability statistics to support the validity of the questionnaire in this adolescent sample.
Response:
We have added lines 587-589 on page 11, page 13, table 2 (Cronbach’s α)
Comment 4. Moderator Analysis Underutilized:
VR immersion and realism scores were collected but not effectively integrated into the analysis. It would strengthen the paper to explore whether immersion moderates the relationship between stress and food craving.
Abstract lines 109-114
Statistical Analysis lines 556-561
Section 3.2 -Lines 576-583
Table 8, 9,10
Discussion section 1092-1098
Comment 5. Causal Language:
Please revise instances where correlational results are interpreted causally (e.g., “predicted” vs. “associated with”). This is important to avoid overinterpretation of cross-sectional or quasi-experimental data.
Response:
Expressions such as moderated, elicits, predicted were modified in the discussion section. (section 4 page 17)
Comment 6. Figures and Visual Aids:
While tables are comprehensive, the manuscript would benefit from visual representations (e.g., ECG changes over time, interaction plots for UE and food type). Consider adding figures to enhance readability and impact.
Response:
We thank you for this helpful suggestion. In response, we have added Figure 4 to visually illustrate the interaction between food type and Uncontrolled Eating (UE) on food desire. This figure enhances the readability of the results and supports the statistical findings reported in Table 6 (ANOVA within-subjects effects).
Comment 7. Discussion Depth:
The discussion could be improved by comparing findings more thoroughly with existing VR-based eating behavior studies and addressing potential mechanisms (e.g., autonomic regulation, reward processing) underlying the observed effects.
Response:
Thank you for your valuable observation. We have reviewed and modified the discussion section. Page 21, lines 1149-1155, lines 1158-1165.
Reviewer 3 Report
Comments and Suggestions for Authors
The manuscript under review, ‘Virtual Reality Trier Social Stress and Virtual Supermarket Exposure: ECG Correlates of Food Craving and Eating Traits in Adolescents,’ is undoubtedly an original and interesting work. In my opinion, the research methods used are correct and the interpretation of the results is accurate.
It is not sufficiently clear to me why the authors only provide data on the number and percentage of individuals in each body weight group in Table 1, without providing detailed characteristics in this regard.
Furthermore, the research group is so small that it would be worth noting in the summary chapter that the research results and conclusions presented should be confirmed on a larger population.
The abbreviation ECG should not be used in the title of the scientific paper, but rather the full name.
Overall, I evaluate the manuscript positively and believe that after making the indicated corrections, it can be accepted for publication.
Author Response
Comment 1.
It is not sufficiently clear to me why the authors only provide data on the number and percentage of individuals in each body weight group in Table 1, without providing detailed characteristics in this regard.
Response:
We thank you for this helpful suggestion. We have modified Table 1, page 11.
Comment 2.
We appreciate your feedback. The sample size was determined a priori using GPower 3.1, based on an expected medium effect size and power, which indicated a required minimum of 34 participants. Our final sample (N = 38). Lines 532-538.
We fully acknowledge that the relatively small sample size may limit the generalizability of the findings. This aspect has been explicitly addressed in the Limitations section, where we emphasize that future studies should replicate and extend these results in larger and more diverse adolescent populations. Page 21, lines 1186-1188.
Comment 3
The abbreviation ECG should not be used in the title of the scientific paper, but rather the full name.
Overall, I evaluate the manuscript positively and believe that after making the indicated corrections, it can be accepted for publication.
Response:
We thank the reviewer for this helpful suggestion. The title has been revised accordingly to replace the abbreviation with the full term: “Virtual Reality Trier Social Stress and Virtual Supermarket Exposure: Electrocardiogram Correlates of Food Craving and Eating Traits in Adolescents.”
Reviewer 4 Report
Comments and Suggestions for Authors
Page 3, lines 159–168: The rationale for combining TSST-VR with food choice paradigms is mentioned but the specific knowledge gap is not clearly defined. Please explicitly state what has not been studied previously.
Page 3, lines 140–157: Several prior studies using VR and TSST are cited, yet the manuscript does not clearly articulate how this study differs mechanistically or methodologically.
Recommendation: Introduce a final paragraph in the introduction that clearly states the novelty, theoretical contribution, and clinical implications of your approach.
Page 4, lines 182–188: The inclusion criteria are clear, but important confounders such as menstrual cycle phase in adolescent females (68% of the sample) are not controlled. This may significantly impact heart rate and craving.
Page 5, lines 197–202: Please clarify whether participants were instructed to fast or avoid caffeine prior to ECG and craving assessment. Without standardization, physiological variability may be due to external factors rather than stress induction.
Page 9, lines 374–375: The manuscript explicitly states that portions of the text were drafted with ChatGPT. While transparency is appreciated, this raises questions regarding originality and scientific accountability. Please provide assurances that no intellectual content, interpretation, or data analysis was generated by AI, and clarify compliance with the journal’s AI usage policy.
Page 10, lines 359–363: No power calculation or effect size justification is provided. Even exploratory studies in journals require a minimum detectable effect size or rationale for sample adequacy.
Recommendation: Include a power analysis or acknowledge this as a limitation with justification of the chosen sample based on prior VR-TSST studies.
Page 6, lines 251–260: Please clarify whether the VR environments were standardized across participants or if any adaptation occurred. If realism scores varied significantly, these should be included as covariates.
Page 8, lines 279–294: The description of ECG signal processing is detailed, but there is no reference to validation in adolescent populations. Please provide reliability metrics or cite pediatric validation studies for the WIWE device.
Page 9, lines 297–301: QTc was calculated using the Fridericia formula; however, QTc norms in adolescents vary by age and sex. Please indicate the clinical thresholds used to interpret QTc changes and justify their relevance to eating behavior.
Page 11, lines 381–388: Statistical significance is reported for PQ interval changes (p = 0.02), but no effect size or confidence interval is provided. This is essential for understanding clinical significance.
Page 12, Table 2: The table includes multiple variables with no clear designation of primary vs. secondary outcomes. Please identify the main outcomes to guide interpretation.
Page 14, lines 447–457: The discussion draws strong conclusions about stress-induced eating behavior; however, causality cannot be inferred from correlational and regression analyses in a small exploratory sample.
Page 16, lines 592–598: The limitations section is too brief. Key limitations missing include:
- Small sample size and potential overfitting of regression models.
- Absence of hormonal cycle control in female participants.
- Reliance on self-reported craving rather than objective intake.
- Potential novelty or gaming effect in VR supermarket, influencing behavior beyond stress mechanisms.
Comments on the Quality of English Language
Several sentences are overly long or repetitive. I recommend a professional language revision to improve clarity and ensure consistency of tense and terminology.
Some references are incomplete or missing DOIs (please check formatting).
Author Response
Comment 1. Page 3, lines 159–168: The rationale for combining TSST-VR with food choice paradigms is mentioned, but the specific knowledge gap is not clearly defined. Please explicitly state what has not been studied previously.
Response:
Thank you for the useful feedback. We have added information on page 4, lines 263-266 and lines 273-275.
Comment 2. Page 3, lines 140–157: Several prior studies using VR and TSST are cited, yet the manuscript does not clearly articulate how this study differs mechanistically or methodologically.
Recommendation: Introduce a final paragraph in the introduction that clearly states the novelty, theoretical contribution, and clinical implications of your approach.
Response:
Page 5, lines 304-315.
Comment 3. Page 4, lines 182–188: The inclusion criteria are clear, but important confounders such as menstrual cycle phase in adolescent females (68% of the sample) are not controlled. This may significantly impact heart rate and craving.
Response:
We have modified, adding lines 324-325.
Comment 4. Page 5, lines 197–202: Please clarify whether participants were instructed to fast or avoid caffeine prior to ECG and craving assessment. Without standardization, physiological variability may be due to external factors rather than stress induction.
Response:
Thank you for this helpful observation. We added lines 331-332.
Comment 5. Page 9, lines 374–375: The manuscript explicitly states that portions of the text were drafted with ChatGPT. While transparency is appreciated, this raises questions regarding originality and scientific accountability. Please provide assurances that no intellectual content, interpretation, or data analysis was generated by AI, and clarify compliance with the journal’s AI usage policy.
Response:
We clarified on page 11, lines 562-565.
Comment 6. Page 10, lines 359–363: No power calculation or effect size justification is provided. Even exploratory studies in journals require a minimum detectable effect size or rationale for sample adequacy.
Recommendation: Include a power analysis or acknowledge this as a limitation with justification of the chosen sample based on prior VR-TSST studies.
Response:
A new section was added on page 10, lines 532-538.
Comment 7. Page 6, lines 251–260: Please clarify whether the VR environments were standardized across participants or if any adaptation occurred.
Response
Thank you for this helpful observation. Clarifications were made on page 6, lines 385-389.
Comment
If realism scores varied significantly, these should be included as covariates.
Response
We thank you for this valuable suggestion. Given the relatively small sample size, adding additional covariates would substantially reduce the statistical power of the analyses and increase the risk of model overfitting. However, to address this, we have added it in the limitation section.
Comment 8. Page 8, lines 279–294: The description of ECG signal processing is detailed, but there is no reference to validation in adolescent populations. Please provide reliability metrics or cite pediatric validation studies for the WIWE device.
Response:
We added for clarification on lines 452-4577.
Comment 9. Page 9, lines 297–301: QTc was calculated using the Fridericia formula; however, QTc norms in adolescents vary by age and sex. Please indicate the clinical thresholds used to interpret QTc changes and justify their relevance to eating behavior.
Response:
We have added lines 462-468 on page 9.
In the discussion section, we have added a study reporting QTc and eating behaviour Lines 1122-1126
Comment 10. Page 11, lines 381–388: Statistical significance is reported for PQ interval changes (p = 0.02), but no effect size or confidence interval is provided. This is essential for understanding clinical significance.
Response:
We have added lines 468-471, Table 2 (Cohen’n d (95% CI)) and lines 547.
Comment 11. Page 12, Table 2: The table includes multiple variables with no clear designation of primary vs. secondary outcomes. Please identify the main outcomes to guide interpretation.
Response:
We thank you for this observation. Table 2 has been revised to clearly distinguish primary outcomes (physiological stress markers: QTc, PQ, HR, and perceived stress measures) and secondary outcomes (eating behavior parameters and virtual reality exposure).
Comment 12. Page 14, lines 447–457: The discussion draws strong conclusions about stress-induced eating behavior; however, causality cannot be inferred from correlational and regression analyses in a small exploratory sample.
Response:
Expressions such as moderated, elicits, predicted were modified in the discussion section.
Comment 13. Page 16, lines 592–598: The limitations section is too brief. Key limitations missing include:
Small sample size and potential overfitting of regression models.
Absence of hormonal cycle control in female participants.
Reliance on self-reported craving rather than objective intake.
Potential novelty or gaming effect in VR supermarket, influencing behavior beyond stress mechanisms.
Response:
We thank you for the constructive feedback. We have added on page 17, lines 1181-1193.
Round 2
Reviewer 4 Report
Comments and Suggestions for Authors
Thank you for your detailed and comprehensive responses to the initial review. I appreciate the effort invested in addressing each of the comments raised.
Overall, the revisions substantially improve the clarity, methodological transparency, and scientific justification of the manuscript. You have added important details in the Introduction, Methods, and Discussion sections, as well as expanded the limitations, which strengthens the overall rigor of the study. Most of the major concerns have been satisfactorily resolved.
Overall, the manuscript has improved considerably after the revisions. While some minor stylistic improvement could still help with coherence and fluidity, the scientific content is now more clearly and accurately communicated.
Comments on the Quality of English Language
The quality of English has improved following the revisions, especially in the newly added methodological and discussion content. Some sentences remain overly complex or repetitive, so a light professional language edit is recommended to ensure clarity, correct tense usage, and consistent terminology.
Author Response
We thank the reviewers for their constructive comments regarding our revised manuscript. We appreciate your acknowledgement of the improvements in clarity, methodological transparency, and scientific justification.
In response to your recommendation, we conducted an additional round of stylistic and language editing. Several long or complex sentences in the Introduction and Discussion were shortened or divided for improved readability, coherence, and consistency in terminology and tense use.